# The Anti-Tumorigenic Role of Cannabinoid Receptor 2 in Colon Cancer: A Study in Mice and Humans

**DOI:** 10.3390/ijms24044060

**Published:** 2023-02-17

**Authors:** Jennifer Ana Iden, Bitya Raphael-Mizrahi, Zamzam Awida, Aaron Naim, Dan Zyc, Tamar Liron, Melody Kasher, Gregory Livshits, Marilena Vered, Yankel Gabet

**Affiliations:** 1Department of Anatomy and Anthropology, Sackler Faculty of Medicine, Tel Aviv University, Tel Aviv 69978, Israel; 2Department of Cell and Developmental Biology, Sackler Faculty of Medicine, Tel Aviv University, Tel Aviv 69978, Israel; 3Department of Morphological Studies, Adelson School of Medicine, Ariel University, Ariel 40700, Israel; 4Department of Oral Pathology, Oral Medicine and Maxillofacial Imaging, The Goldschleger School of Dental Medicine, Sackler Faculty of Medicine, Tel Aviv University, Tel Aviv 69978, Israel; 5Institute of Pathology, The Chaim Sheba Medical Center, Tel Hashomer, Ramat Gan 52621, Israel

**Keywords:** cannabinoid receptor 2, endocannabinoid system, immunomodulation, colon cancer, tumor microenvironment, myeloid-derived suppressor cells

## Abstract

The endocannabinoid system, particularly cannabinoid receptor 2 (CB2 in mice and CNR2 in humans), has controversial pathophysiological implications in colon cancer. Here, we investigate the role of CB2 in potentiating the immune response in colon cancer in mice and determine the influence of *CNR2* variants in humans. Comparing wild-type (WT) mice to CB2 knockout (CB2^−/−^) mice, we performed a spontaneous cancer study in aging mice and subsequently used the AOM/DSS model of colitis-associated colorectal cancer and a model for hereditary colon cancer (Apc^Min/+^). Additionally, we analyzed genomic data in a large human population to determine the relationship between *CNR2* variants and colon cancer incidence. Aging CB2^−/−^ mice exhibited a higher incidence of spontaneous precancerous lesions in the colon compared to WT controls. The AOM/DSS-treated CB2^−/−^ and Apc^Min/+^CB2^−/−^ mice experienced aggravated tumorigenesis and enhanced splenic populations of immunosuppressive myeloid-derived suppressor cells along with abated anti-tumor CD8+ T cells. Importantly, corroborative genomic data reveal a significant association between non-synonymous variants of *CNR2* and the incidence of colon cancer in humans. Taken together, the results suggest that endogenous CB2 activation suppresses colon tumorigenesis by shifting the balance towards anti-tumor immune cells in mice and thus portray the prognostic value of *CNR2* variants for colon cancer patients.

## 1. Introduction

According to the National Institute of Health, cancer is among the leading causes of death globally, with the number of new cancer cases per year expected to rise to 23.6 million by 2030 [1]. Specifically, colon carcinoma is one of the most common malignant tumors observed in the clinic and the fourth leading cause of tumor-related deaths [2]. Chronic inflammation is the most significant risk factor for human colorectal cancer (CRC). Patients with inflammatory bowel disease, Crohn’s disease, or ulcerative colitis are at high risk and have a higher mortality rate than other CRC patients [3].

The azoxymethane/dextran sodium sulfate (AOM/DSS)-induced mouse model of colitis-associated colorectal cancer (CAC) is one of the most commonly used chemically induced CRC models due to its reproducibility and potency [4]. Studies based on this model have confirmed the significance of inflammation in CRC development and have illuminated some of the mechanisms of inflammation-related colon carcinogenesis. Specifically, the results have implicated the function of pro- and anti-inflammatory cytokines in CRC pathology and the contribution of an immunosuppressive environment, as well as the roles of specific subsets of immune cells [5,6,7]. 

A number of transgenic mouse models have now been developed to replicate the hereditary factors known to be associated with human colon cancer. The well-established Apc^Min/+^ genetic model spontaneously develops multiple intestinal neoplasia and mirrors human familial adenomatous polyposis and colorectal tumors. These mice carry a mutation in the murine homolog of the *Apc* gene, which is a crucial tumor-suppressor gene belonging to the canonical Wnt signaling pathway. This model has been used previously to establish the tumor-promoting role of myeloid-derived suppressor cells (MDSCs) [8,9,10]. 

MDSCs in CRC patients correlate with reduced survival and have emerged as a major obstacle to cancer immunotherapy [2]. This heterogeneous population accumulates in the circulation and lymphoid organs, and in particular in the bone marrow, and tumor sites of tumor-bearing patients. These MDSCs have been shown to promote invasion, angiogenesis, and tumor immune evasion, leading to sustained cancer progression in a number of experimental animal models of cancer. 

Endocannabinoids and endocannabinoid receptors comprise the endocannabinoid system (ECS) and cooperate to mediate diverse biological and behavioral processes [11]. The endocannabinoid receptors, which are G-protein-coupled receptors (GPCR), are differentially expressed in various tissues where they fulfil manifold functions [12]. The ECS consists of two prominent endogenous ligands, namely N-arachidonoylethanolamine (AEA or anandamide) and 2-arachidonoylglycerol (2-AG), and the cannabinoid receptors CB1 and CB2 (CNR1 and CNR2 in humans) [13]. CB1 is expressed predominantly in neurons, while CB2 is expressed mainly in immune and bone cells [14]. Due to the relatively high expression of CB2 in immune cells, this receptor has been suggested to mediate the immunomodulatory effects of endocannabinoids [15]. The specific role of CB2 in cancer has not been thoroughly investigated in models with an intact immune system, and the question as to whether CB2 protects against or promotes cancer development remains controversial [16,17,18,19,20,21]. This outstanding issue, combined with our preliminary data that suggest CB2 has a protective role in tumorigenesis, provided the motivation to determine the role of endogenous CB2 activation in stimulating the host immune response against tumor development. Here, we tested the hypothesis that CB2 plays a protective role in suppressing tumorigenesis using a systemic CB2 knockout (CB2^−/−^) model and well-established colon cancer models in immunocompetent mice and examined the association between the CNR2 gene variants and colon cancer in a human population.

## 2. Results

### 2.1. CB2^−/−^ Is Associated with an Increased Risk of Spontaneous Cancer

Wild-type (WT) mice grown in SPF conditions do not typically develop cancer in the first year of life. To examine whether CB2 contributes to natural resistance to spontaneous cancer, we performed a histopathological screen of 14-month-old WT and CB2^−/−^ male and female mice. A careful analysis of the tissues collected from the colon and reproductive organs (uterus, ovaries, and testicles) revealed the occurrence of cancerous and pre-cancerous lesions in CB2^−/−^ mice in a tissue- and sex-preferential manner. Female CB2^−/−^ mice demonstrated a significant increase in the incidence of cancerous and precancerous lesions (normal versus abnormal, Fisher’s exact test) and severity (Mann–Whitney U Test) in the colon and uterus (Figure 1A). In contrast, there was no occurrence of precancerous lesions in the colon of either WT or CB2^−/−^ male mice although there was a three-fold higher (although not statistically significant) incidence of precancerous lesions in the testicles of CB2^−/−^ compared to sex-matched WT mice (Figure 1A). These observations indicate that CB2 has a protective role throughout life against tumorigenesis across cancer types, especially in females.

### 2.2. IL-6 Is Upregulated in CB2^−/−^ Female Mice under Steady-State Conditions

The notion that CB2 plays a role in interleukin-6 (IL-6) production has been reported previously [22,23]. IL-6 is upregulated in excessive inflammatory conditions in CB2^−/−^ males [24,25] and has been shown to direct hematopoiesis towards a myeloid output where such abnormal myelopoiesis contributes to the upregulation of MDSCs in cancer and inflammation [26,27]. Because of the sex-related increased incidence of precancerous lesions detected in CB2^−/−^ females, we assessed the levels of IL-6 in the serum of 12–14-week-old male and female WT and CB2^−/−^ mice. The results indicated a significant upregulation of IL-6 in the female knockout mice, although there were no significant differences in IL-6 serum levels between male CB2^−/−^ and WT mice (Figure 2). This corroborates our observation of an increased incidence of spontaneous precancerous lesions in the colon and reproductive organs of only female CB2^−/−^ mice and suggests a role for IL-6 and MDSCs in CB2-related cancer development.

### 2.3. CB2 Has a Protective Role against Carcinogen-Induced Colon Cancer in Female Mice

As a further investigation of the role of CB2 in cancer development, we focused on colon cancer in female mice. Cancer induction started at the age of six weeks (AOM injection) and the mice received three cycles of DSS in their drinking water followed by a recovery period up to 17 weeks of age. Cancer progression was assessed by monitoring weight loss. The CB2^−/−^ mice at the beginning of the experiment (6 weeks old) were slightly but significantly heavier than the age- and sex-matched WT animals (+8.7%, *p* = 0.015, not shown), but during CAC induction, these mice lost significantly more weight (Figure 3A). Importantly, weight loss can be aggressive during DSS administration, and for ethical reasons, mice are sacrificed if they lose more than 10% body weight between a two-day weighing period. Notably, while all the WT mice reached the end of the experiment, 40% of the CB2^−/−^ animals had to be sacrificed early due to significant weight loss (Figure 3B). The results of a colonoscopy after the second DSS cycle in vivo revealed a significantly higher level of disorganization of blood vessels in the CB2^−/−^ mice and an overall higher disease activity index (Figure 3C). The colons in the CB2^−/−^ mice were significantly shorter than in the controls (Figure 3E), an observation that is indicative of the degree of colonic inflammation in DSS-treated animals [21]. Importantly, we observed significantly more polyps in the CB2^−/−^ animals than in the WT mice (Figure 3D). Accordingly, our histopathological analysis indicated significantly higher pathology scoring with an increased number of dysplastic polyps in the CB2^−/−^ mice vs. WT (Figure 3F,G).

### 2.4. CB2 Has a Protective Role against Colon Cancer in Apc^Min/+^ Mice

As an alternative approach, we considered a genetic model of colon cancer based on Human familial adenomatous polyposis, which is caused by loss of function mutations in the Apc gene [28]. Mice lacking one Apc allele (Apc^Min/+^) develop colon cancer with no further induction and usually die at 6 months of age [28]. Initially, we compared the survival of Apc^Min/+^CB2^+/+^ to Apc^Min/+^CB2^−/−^ mice and found that the absence of CB2 reduces 6-month survival by 35% for males and females combined (*p* = 0.016, Figure 4A). To assess the tumor-suppressive role of endogenous CB2 activation in this model, we compared Apc^Min/+^CB2^+/+^ mice to Apc^Min/+^CB2^−/−^ mice at 9 weeks of age. Because we found no differences in cancer development and survival between male and female mice, we combined the results from both sexes. Like the AOM/DSS model, cancer development in the CB2-deficient Apc^+/Min^ mice appeared more aggressive (Figure 4), with shorter colons (Figure 4D) and an increased number of adenomas in the small and large intestine (SI and LI, respectively, Figure 4B,C) seen in the Apc^Min/+^CB2^−/−^ mice. 

### 2.5. Splenic Immunosuppressive Profile in CAC-Induced CB2^−/−^ and Apc^Min/+^ CB2^−/−^ Mice 

Based on previous reports that have attributed immunosuppressive and tumor-promoting properties to M-MDSCs and PMN-MDSCs, we evaluated the presence of these populations in the spleen in both colon cancer models [29,30]. Flow cytometry revealed a significantly higher number of these cells in the spleens of CB2-deficient female mice receiving AOM/DSS (Figure 5C). In addition, the side/forward-scatter of the cells from CAC-induced CB2^−/−^ mice revealed a strikingly different immune profile overall with a higher number of myeloid cells relative to the appropriate control animals (CAC-induced WT). Specifically, the number of myeloid CD11b+ cells was significantly increased in the CB2-deficient mice (Figure 5B). This is of particular importance because CD11b upregulation has been positively correlated with cancer progression across multiple models of cancer [31,32]. The Apc^Min/+^ CB2^−/−^ mice exhibited a similar immunosuppressive profile, with upregulation of CD11b+, M-MDSCs, and PMN-MDSCs (Figure 6A), although in contrast to the AOM/DSS model, dendritic cells (DCs) were also upregulated, while anti-tumor eosinophils and CD8+ T cells were downregulated (Figure 6A). Notably, CD4+ T cells were relatively upregulated in the absence of CB2, with an altered CD4:CD8 ratio in the Apc^Min/+^ mice (Figure 6B), while the CB2^−/−^ mice in the CAC model had a significantly lower percent of CD3+ T cells, with lower numbers of both CD4+ and CD8+ T cells (Figure 5A). Interestingly, there were no marked differences between the genotypes in either cancer model in the number of macrophages (Figure 5C and Figure 6A).

### 2.6. Immunosuppressive Environment in Colon Polpys of CB2^−/−^ Mice

As PMN-MDSCs were upregulated in the spleens of mice lacking the CB2 receptor, we assessed the presence of these tumor-supporting myeloid cells in the tumor microenvironment (TME) of mice treated with AOM/DSS. We found that this immunosuppressive population is significantly higher in CB2^−/−^ mice (Figure 7B). Because IL-6 is a crucial regulator of MDSC activity and proliferation, leading to tumor cell survival, coupled with upregulated IL-6 in the serum of naïve CB2^−/−^ females (Figure 2), we assessed the mRNA expression of IL-6 in the TME along with arginase-1 (Arg1), a functional marker of MDSCs. IL-6 and Arg1 were both significantly upregulated in the polpys of CB2^−/−^ mice (Figure 7A). While IL-17 is known to have double-edged features—where in early tumor development, IL-17 promotes tumor growth, while in later stages, it has been shown to enhance cytotoxic T cells (CD8+) to generate an anti-tumor effect and is positively correlated with survival in human adenocarcinoma [33]. We assessed the levels of IL-17 and CD8 and found that they are significantly downregulated in mice lacking the CB2 receptor (Figure 7A). Nitric oxide (NO) is known to be secreted by MDSCs and is a critical factor in T cell suppression [34]. NO levels in the supernatant of the tumor fragments of polyps excised from the distal colon of AOM/DSS-treated mice were significantly higher in the CB2^−/−^ mice (Figure 7C). This finding supports the notion that the higher numbers of MDSCs and increased immunosuppressive secretory factors (NO) in the CB2 knockout mice systemically and in the TME contribute to T cell suppression and increased cancer severity. 

### 2.7. Association of Polymorphisms in the CNR2 Gene with Colon Cancer Incidence in Humans

The genomic region of interest on chromosome 1 between base pairs 23,920,590 and 25,516,845 includes 287 single nucleotide polymorphisms (SNPs), of which 63 neighboring SNPs mapped to CNR2 revealed statistically significant [*p* < 0.05 (log_(10)_ < −1.3] associations with colon cancer. All of these SNPs were mapped to a single haploblock (Figure 8A) with *p*-values ranging from *p* = 0.0184 to *p* = 0.0272 (Figure 8B). The LDs between each pair of tested SNPs as measured by r^2^ were uniformly close to 1.0, suggesting close linkage between all 63 SNPs. Interestingly, seven synonymous SNPs in the exon 3 region of CNR2, and a nonsynonymous mutation could be detected. This non-synonymous SNP, rs2501432, exhibited a significant association with *p* = 0.025. This polymorphism is in strong LD (r^2^ = 1) with the other 62 significant CNR2 SNPs. Notably, another SNP (rs151243307) that is located outside of the CNR2 gene but within the haplotype block also appears to be highly significantly associated (4.37 × 10^−9^) with the phenotype of the study, and the LD was high with all 63 SNPs (r^2^ = 1 for all, Appendix A).

## 3. Discussion

Colon cancer is complex, and the exact cause is undetermined, but an imbalance in immune cells as well as a genetic predisposition are known to be contributory factors in disease development. As corroborated by our murine data, colon cancer exhibits sex-related incidence, severity, and underlying mechanisms [35]. 

The results of our study demonstrate that CB2 protects against the development of colon cancer in two different models. This was reflected by a higher incidence of spontaneous precancerous lesions in aging CB2^−/−^ female mice, accompanied by a predisposition to other types of cancer. In addition, disease severity and the number of tumors were higher in a chemically induced colon cancer model in female mice lacking CB2. The TME of CB2^−/−^ mice showed higher levels of immunosuppressive cells and secretory factors (NO) accompanied by downregulated levels of pro-inflammatory IL-17. The results with a model relevant to genetically induced colon cancer, indicate that CB2 protects both male and female mice against colon cancer progression. A consideration of possible mechanisms suggested that Apc^Min/+^CB2^−/−^ mice possess an increase in immunosuppressive and tumor-promoting cells in the spleen, associated with a decrease in anti-tumor CD8+ T cells and eosinophils (compared to the Apc^Min/+^CB2^+/+^ controls). Taken together, these data suggest that endogenous CB2 activation suppresses colon cancer development by altering the balance between pro-tumorigenic and anti-tumorigenic immune cells in the spleen, and by reducing the levels of immunosuppressive factors in the tumor-microenvironment. 

Importantly, clinical data also support the crucial nature of the role of the CB2 receptor in colon cancer development. Indeed, corroborative data from a large human population databank (UK Biobank) reveal a significant association between non-synonymous variants of *CNR2* and colon cancer incidence in humans. Of particular interest is the finding that a similar genomic region has been implicated in two independent reports that demonstrated an association between *CNR2* and bone mineral density in humans [36,37]. Osteoporosis and cancer are both progressive, aging-associated diseases and CNR2/CB2 activity protects against both diseases. This challenges the idea that anandamide and 2-AG, two endogenous cannabinoids that are released on-demand and rapidly degraded [38], can maintain the assumed long-term CB2 tone. In this context, we recently reported the occurrence of a third endogenous CB2-selective agonist named the osteogenic growth peptide (OGP) [39], which is highly conserved among mammals, and is physiologically present in the serum at nano- to micromolar concentrations [40,41,42]. Interestingly, the physiological levels of OGP decrease with age in humans, and one may assume that this results in an age-related decline in CB2 tone, thus reducing the protection provided by CB2 against both osteoporosis and cancer. 

Much of the focus of intestinal cancer models has been on investigating how MDSCs promote tumor growth, in part via NO production that suppresses T cells. Based on the tendency of the CB2^−/−^ mice to develop spontaneous colon cancer, we hypothesized that high IL-6 in the periphery leads to upregulation of MDSCs [27]. CB2 may act by modulating the levels of IL-6 under low inflammatory conditions, while CB2^−/−^ females, which have increased IL-6 levels (Figure 2) even under non-inflammatory conditions, are at a higher risk for colon cancer in the steady-state because of the IL-6-induced upregulation of MDSCs. This is opposed to CB2^−/−^ males, who show no difference in IL-6 levels at steady-state compared to WT. CB2 agonists have been shown to modulate the levels of sex hormones such as progesterone and estradiol, which in turn can affect IL-6 and myeloid cell proliferation [43,44]. In addition, sex differences in the response to exogenous cannabinoids appear to be strongly influenced by ovarian hormones [45]. This possibly explains the sex disparities seen in the spontaneous cancer model, and when coupled with our results, suggest that sex differences in endocannabinoid tone determine differential sensitivity and susceptibility to tumorigenesis. In contrast, under highly inflammatory conditions CB2 knockout mice and humans with reduced CNR2 function display higher levels of IL-6 independently of sex [24,25,46,47]. The observation that IL-6 is strongly upregulated in the inflammatory environment of Apc^Min/+^ mice [48,49] and colon cancer patients in males and females [50,51,52] may therefore explain the lack of sex-related differences detected in the influence of CB2 on colon cancer severity in our Apc^Min/+^ model (Figure 4). 

CB2 has previously been shown to modulate the migration of myeloid cells towards a site of inflammation, and the absence of CB2 enhances myeloid cell recruitment towards the spleen [53]. Although we did not detect any CB2-related differences in the number of macrophages in the spleen, we cannot rule out their possible involvement in the anti-tumorigenic role of CB2, e.g., by polarizing macrophages towards M1 macrophages (tumor promoting) or by limiting the expression of IL-6 and other chemokines [54,55,56]. Furthermore, we cannot negate the role of the CB2 receptor in the formation and regulation of T and B cell subsets [57,58]. There have been some reports that CB2 agonists can reduce colonic inflammation by attenuating T cell activity, while others present evidence for the role of CB2 activation as a mediator between myeloid and T cells [59,60].

While our data support the notion that activation of CB2 on immune cells is anti-tumorigenic, others have implicated the expression of CB2 on gut epithelial or endothelial cells, including cross-talk between CB2 and PPARγ (expressed on epithelial cells and macrophages), as well as cross-talk mediated by CB2 between cancer cells and endothelial cells, as a potential mechanism for the pro- or anti-tumorigenic effect [61,62]. As already described, CB2 has been investigated in multiple cancer types and models of inflammation, and there are controversial results regarding the effect on tumor progression: CB2 expression is associated with a poor prognosis in humans, CB2 antagonists suppress tumorigenesis, CB2 activation promotes tumorigenesis in models of colon cancer, CB2 agonists inhibit carcinogenesis, etc. [16,17,18,21,63,64,65]. Importantly, many of these studies used xenograft models with immortalized cancer cell lines, whose ability to evade the immune system is well established. Our use of chemical or genetic induction to evaluate the role of CB2 in potentiating the immune response may explain any disparities between previous reports and our results. An alternative explanation for paradoxical results, particularly with CB2 synthetic agonists and antagonists, includes biased signaling and functional selectivity via CB2, which activates multiple pathways, and the observation that certain preferential signaling cascades are tissue specific [66,67,68,69]. 

In conclusion, our results in both chemical and genetic models of colon cancer demonstrate that endogenous CB2 activation can modulate the immune response and consequently reduce tumorigenesis. This implies that CB2 could have an anti-tumorigenic role in colon cancer and serve as a target in personalized medicine. 

## 4. Materials and Methods

### 4.1. Mice

Mice on a C57Bl/6J genetic background were bred in a specific pathogen-free (SPF) animal facility at Tel Aviv University. All of the mice were housed as per IACUC guidelines in temperature-controlled rooms with a 12 h light cycle and were given water and pelleted chow ad libitum. All of the experiments were conducted in accordance with the guidelines and with the approval of the Tel Aviv University Animal Care and Use Committee (Protocols 01-18-059 and 01-21-010). For the spontaneous cancer study, male and female WT and CB2^−/−^ mice (provided by Andreas Zimmer), eight per group, were kept under normal SPF conditions and tissues were collected from the colon, skin, liver, and reproductive organs (uterus, ovaries, and testicles) at the age of 14 months or sooner if cancerous lesions were observed. In the induced cancer model, 6-week-old female WT and CB2^−/−^ mice, ten per group, were given a single intraperitoneal injection of AOM (12 mg/kg body weight). Three days after the AOM injection, the mice were given 2.5% DSS in their drinking water for 7 days in weeks 1, 4, and 7. Cancer progression in all of the mice was monitored through body weight, fecal occult blood, and colonoscopy. After 11 weeks, the mice were anesthetized with ketamine (80 mg/kg) and xylazine (12 mg/kg) followed by cervical dislocation and organ harvest. The Apc^Min/+^ (heterozygous) mice were obtained from the Jackson Laboratory were subsequently bred in-house using heterozygous males and WT females and were maintained as heterozygous mutants on a C57BL/6J background. Apc^Min/+^ heterozygous males were crossed with CB2^−/−^ females to generate Apc^Min/+^ CB2^+/−^ males who were then crossed with CB2^−/−^ females to generate Apc^Min/+^ CB2^−/−^ mice. The Apc^Min/+^ CB2^−/−^ were maintained as heterozygous mutants by breeding Apc^Min/+^ CB2^−/−^ males with CB2^−/−^ females. For the Apc^Min/+^ survival experiments, 12 Apc^Min/+^ CB2^+/+^ (6 females and 6 males) and 20 Apc^Min/+^ CB2^−/−^ (8 females and 12 males) mice were examined biweekly and were sacrificed upon the appearance of a moderate rectal prolapse (>1 mm), a significant change in bowel movement (e.g., bloody stool), or stress-related behavior. For subsequent tumor count experiments, 16 Apc^Min/+^ CB2^+/+^ (8 females and 8 males) and 23 Apc^Min/+^ CB2^−/−^ (10 females and 13 males) mice were used, and of those, 11 Apc^Min/+^ CB2^+/+^ (5 females and 6 males) and 11 Apc^Min/+^ CB2^−/−^ (5 females and 6 males) were used for flow cytometry analysis. 

### 4.2. Genotyping

Genomic DNA was extracted from tail clippings and Extracta DNA Prep for PCR (Quantabio, Beverly, MA, USA). The PCR was performed with a DreamTaq Green PCR master mix (Thermo Scientific). The following primers were used: Apc^Min/+^ wild-type forward, 5′-GCCATCCCTTCACGTTAG-3′,Apc^Min/+^forward, 5′-TTCTGAGAAAGACAGAAGTTA-3′, and Apc^Min/+^ common antisense, 5′-TTCCACTTTGGCATAAGGC-3′ for Apc^Min/+^. CB2^−/−^: 5′-AGCGCATGCTCCAGACTGCCT-3′ AGCGCATGCTCCAGACTGCCT, CB2^+/+^ 5′-GTGCTGGGCAGCAGAGCGAATC-3′, and CB2 common antisense: 5′-GTCGACTCCAACGCTATCTTC-3′. 

### 4.3. Serum Analysis

Interleukin-6 (IL-6) levels in the serum of 12–14-week-old mice were measured using a murine IL-6 pre-coated ELISA kit (Peprotech, Rehovot, Israel) according to the manufacturer’s instructions.

### 4.4. Histopathology

All of the pathological analyses were performed by a board-certified pathologist (M.V.). Abnormal tissue or cells were graded according to the following scale: 0—normal tissue; 1—chronic inflammation, ulceration, atrophy, or hyperplasia; 2—cellular atypia, metaplasia, or mitotic figures; 3—dysplasia; 4—positive carcinoma. Cancer severity as well as the occurrence of pre-cancerous and cancerous lesions in the CB2^−/−^ mice were compared to the situation in the WT.

### 4.5. Flow Cytometry 

Single-cell suspensions were obtained by manually homogenizing the harvested spleens in a Petri dish. Red blood cells were lysed using ACK Lysis buffer (Life Technologies, Chicago, IL, USA). The cells were washed with PBS, filtered through a 70 μm cell strainer (Falcon, Corning Incorporated) and then resuspended in PBS. The T cells were stained with anti-CD3-FITC, anti-CD4-PE, and anti-CD8-APC for 30 min on ice, while the myeloid cells were stained with anti-CD45-Pacific Blue, anti-CD11b-PE-Cy7, anti-CD11c-PE, anti-Ly6G-FITC, anti-Ly6C-PerCP-Cy5.5, and anti-Siglec-F-APC or anti-F4/80-APC for 45 min on ice. All of the antibodies were purchased from BioLegend. After staining, the cells were washed twice with PBS and the fluorescence was assessed with a CytoFlex5L (Beckman Coulter, Brea, CA, USA). The dendritic cells were classified as CD45+CD11b+CD11c+, the macrophages were classified as CD45+CD11b+F4/80+, the eosinophils were classified as CD45+CD11b+Siglec-F+, PMN-MDSCs were classified as CD45+CD11b+CD11c^lo/neg^Ly6G^hi^Ly6C^int^, and M-MDSCs were classified as CD45+CD11b+CD11c^lo/neg^Ly6G-Ly6C^hi^. Analysis was performed using CytExpert^®^ (Beckman Coulter, Brea, USA). For the gating strategy of the MDSCs in the spleen, see Appendix A. 

### 4.6. Colonoscopies

Colonoscopies were performed at week 6 after cancer induction by a certified veterinarian (Dr. Mickey Harlev, Tel Aviv University) and scored as described by Becker et al. [19]. Briefly, disease activity scores ranging from 0–2 were given, for the transparency of the colonic wall, the looseness of feces, blood vessel organization, granulation, fibrin deposition, and weight loss.

### 4.7. Colon Histology

The colons were flushed with PBS immediately after harvesting, followed by fixation in formalin in a Swiss roll formation for at least 24 h. The specimens were then paraffin-embedded, sliced into 5 µm sections, and stained with hematoxylin–eosin (H&E) according to Bialkowska et al. [20]. Due to the intensive colon damage caused by harvesting the tumors for flow cytometry and the nitric oxide assays, three WT and four CB2^−/−^ mice receiving AOM/DSS were dedicated to histopathological analyses. 

### 4.8. RNA Extraction and qPCR

Total RNA was extracted from the polyps of similar size from the distal colon of each mouse using TRIzol reagent (Invitrogen, Carlsbad, CA, USA), and qPCR was performed using cDNA generated from 1 µg of total RNA with a cDNA synthesis kit (Quantabio, Beverly, MA, USA). qPCR reactions were carried out on 20 ng cDNA per reaction using n SYBR Green PCR master mix (Quantabio, Beverly, MA, USA) using a step-one (Thermo Fisher, Waltham, MA, USA) analysis system. Relative expression values were quantitated using the comparative cycle threshold method and normalized to mouse β-actin. The following primers were used:

Primer Sequence 5′-3′β-actin_FGTCACCCACACTGTGCCCATCβ-actin_RCCGTCAGGCAGCTCATAGCTCIL-6_FCCGGAGAGGAGACTTCACAGIL-6_RGGAAATTGGGGTAGGAAGGAArg1_FTTGGGTGGATGCTCACACTGArg1_RTTGCCCATGCAGATTCCCIL-17_FACCGCAATGAAGACCCTGATIL-17_RTCCCTCCGCATTGACACACD8_FCCGTTGACCCGCTTTCTGTCD8_RCGGCGTCCATTTTCTTTGGAA

### 4.9. Nitric Oxide (NO) Assessment in Colon Tumor Fragments

Distal colon tumors were weighed, cut into 1 mm fragments, and cultured in a normalized volume of RPMI supplemented with 10% FBS (based on tumor weight) for 8 h. After this time, the NO in the supernatant was measured by a Griess Reagent System (Promega, Madison, WI, USA) according to the manufacturer’s instructions. 

### 4.10. Statistical Analyses

All of the analyses were conducted using GraphPad Prism v9.0. The data were analyzed by the Student’s t-test or the Mann–Whitney U test for continuous variables and the Fisher’s exact test for categorical variables. Differences in weight loss were analyzed by two-way ANOVA for repeated measures over time. All data are represented as mean ± SD unless otherwise stated. The survival curves were analyzed using the log-rank (Mantel–Cox) test. 

### 4.11. Human Genomic Data

We used the GWAS summary statistics for colorectal cancer from the UK Biobank, prepared by the Neale Lab [70]. The summary statistics file contains 510 cases and 360,631 controls and includes both males and females. In the present analysis, we were particularly interested in examining the haplotype block that contains the gene of interest, *CNR2*. This region was identified using *ldetect* [71] and was mapped to chromosome 1p36.11 where it spans base pairs 23,920,590–25,516,845, with the *CNR2* gene located between base pairs 24,197,005 and 24,239,852. Thus, the entire gene with all the available SNPs for the present analysis are encompassed in the single haploblock with the LD ranging from 0.001 to 1.00 as estimated by r^2^ measure. In total, 287 SNPs covering the entire haploblock were available for analysis. We computed the LD matrix in our sample using *LDlink*, which generates the matrix and r^2^ estimates [72]. Following the identification of the *CNR2*-containing genomic region, a gene enrichment analysis was conducted using the Functional Mapping and Annotation (FUMA) GWAS platform to confirm the presence of the *CNR2* gene and its association with each SNP that showed a *p*-value < 0.05 in the targeted region. The *manhattan* function in the *qqman* R statistical package was used to identify the orientation of the SNPs. The generated plot describes the location of each SNP and its corresponding *p*-value following a logarithmic transformation [73].

## Figures and Tables

**Figure 1 ijms-24-04060-f001:**
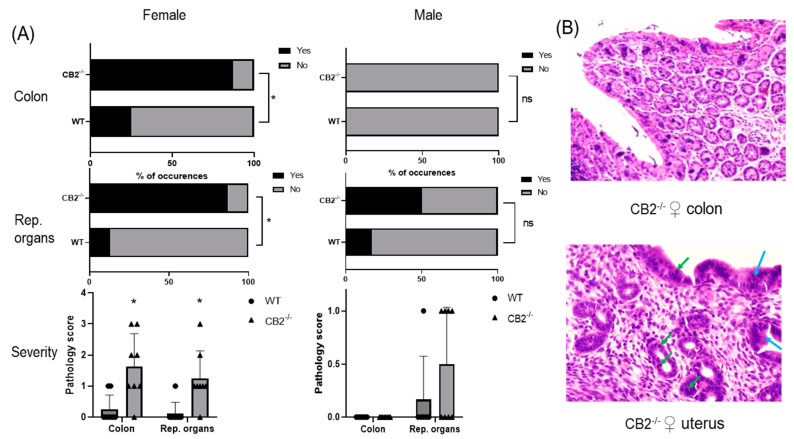
CB2^−/−^ is associated with an increased risk of cancer in aged male and female mice. (**A**) Increased odd’s ratio in CB2^−/−^ mice to develop cancerous and pre-cancerous lesions (“yes” occurrences) in the colon and reproductive organs (rep. organs) as assessed histopathologically; Fisher’s exact test, * *p* < 0.05 versus WT mice; ns, not significant. Severity of dysplasia (bottom row) according to the scale: 0—normal tissue; 1—chronic inflammation, ulceration, atrophy, or hyperplasia; 2—cellular atypia, metaplasia or mitotic figures; 3—dysplasia; 4—positive carcinoma. (**B**) Severe dysplasia and carcinoma in situ in the colon of a CB2^−/−^ female (score of 3, top row). Mitotic figures (green arrows) and dysplasia (blue arrows) in female CB2^−/−^ uterus (score of 3, bottom row). Original magnification ×40. Mann–Whitney U test * *p* < 0.05.

**Figure 2 ijms-24-04060-f002:**
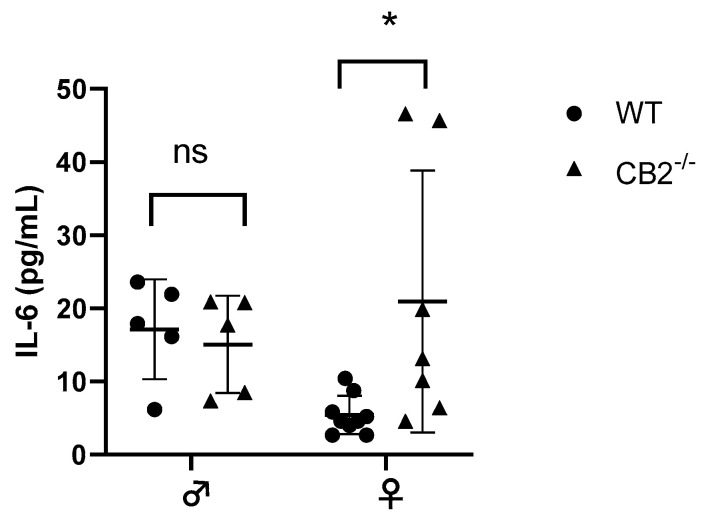
IL-6 serum levels of wild-type (WT) and CB2^−/−^ male and female mice at 12–14 weeks of age. Serum IL-6 was analyzed with the ELISA technique. n ≥ 5; Student’s *t*-test, * *p* < 0.05; ns: not significant.

**Figure 3 ijms-24-04060-f003:**
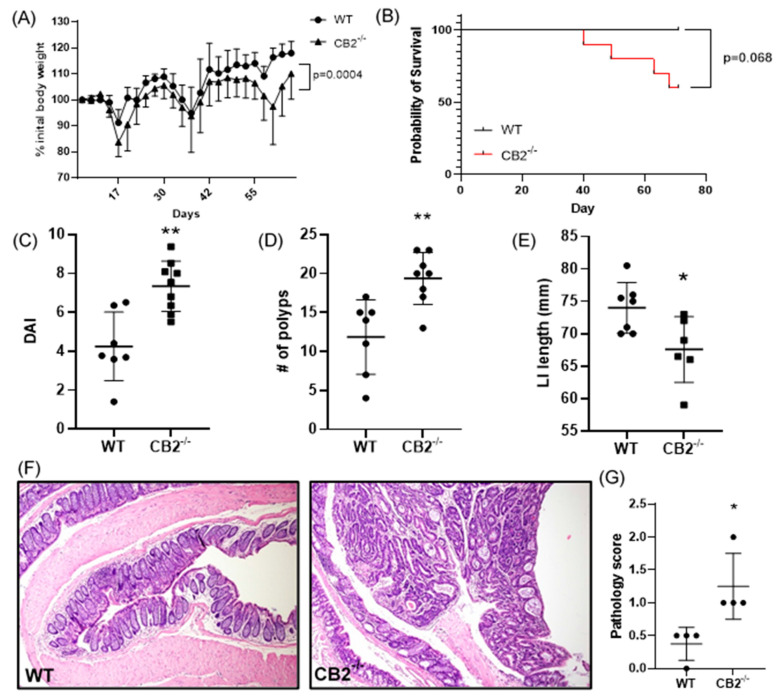
The CB2 receptor protects against CRC development in the AOM/DSS model. (**A**) Body weight expressed as percent (%) of initial weight on day 0; only mice that survived the entirety of the experiment were included, two-way ANOVA. (**B**) Survival curve, Mantel Cox test; (**C**) disease activity index (DAI) as measured by colonoscopy at week 7; this includes mice that did not survive the entirety of the experiment (WT, n = 7; CB2^−/−^, n = 9). (**D**) Number of polyps in the large intestine; this includes two CB2^−/−^mice that died two days before the conclusion of the experiment (WT, n = 7; CB2^−/−^, n = 8). (**E**) Length of the large intestine (LI). (**F**) Representative histological images of WT (left) and CB2^−/−^ (right) colons. Original magnification ×40. (**G**) Pathological score graded according to the following scale: 0—normal tissue; 1—chronic inflammation, ulceration, atrophy, or hyperplasia; 2—cellular atypia, metaplasia or mitotic figures; 3—dysplasia; 4—positive carcinoma. Mann–Whitney U test or Student’s t-test unless otherwise stated. WT, n ≥ 4; CB2^−/−^, n ≥ 4 as indicated. * *p* < 0.05, ** *p* < 0.01.

**Figure 4 ijms-24-04060-f004:**
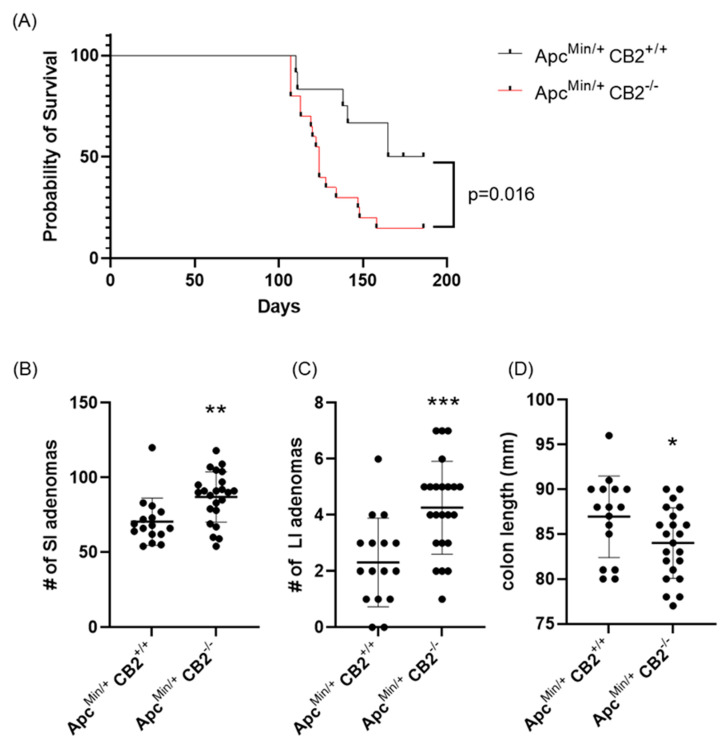
The CB2 receptor protects against cancer progression in the Apc^Min/+^ male and female mice. (**A**) Survival curve, (**B**) number of adenomas in the small (SI) and (**C**) large intestine (LI), and (**D**) colon length. Apc^Min/+^ CB2^+/+^, n = 12 (6 females and 6 males); Apc^Min/+^ CB2^−/−^, n = 20 (8 females and 12 males), Kaplan–Meier (survival curve), Apc^Min/+^ CB2^+/+^, n = 16 (8 females and 8 males); Apc^Min/+^ CB2^−/−^, n = 23 (10 females and 13 males), Mann–Whitney U test or Student’s *t*-test, * *p* < 0.05, ** *p* < 0.01, *** *p* < 0.001.

**Figure 5 ijms-24-04060-f005:**
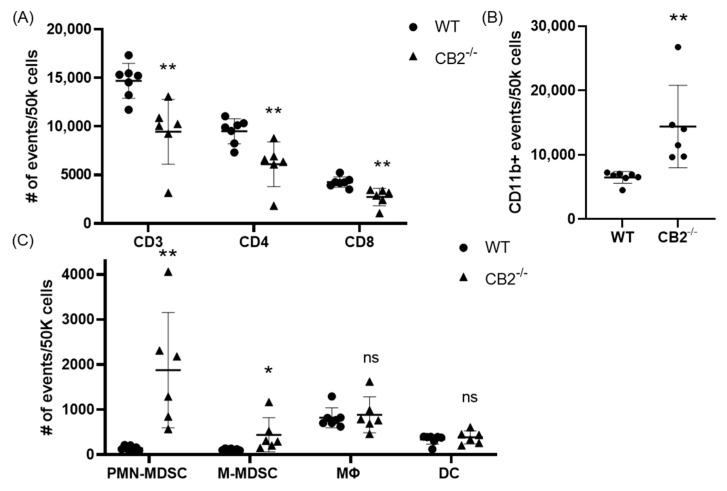
The CB2 receptor alters the balance between pro-tumor and anti-tumor cells in the spleen in mice receiving AOM/DSS. (**A**) Relative frequency of CD3+, CD3+CD4+, and CD3+CD8+ T cells in the spleen. (**B**) Relative frequency of CD11b+ and subpopulations of (**C**) CD11b+Ly6G^hi^Ly6C^int^ (PMN-MDSC), CD11b+Ly6G-Ly6C^hi^ (M-MDSC), CD11b+F480+ (macrophage, MΦ), and CD11b^hi^ Cd11c^hi^ (DC). All of the values are expressed as the number of events per 50,000 cells, determined by flow cytometry. WT, n ≥ 7; CB2^−/−^, n = 6. Student’s *t*-test, * *p* < 0.05 and ** *p* < 0.01; ns: not significant.

**Figure 6 ijms-24-04060-f006:**
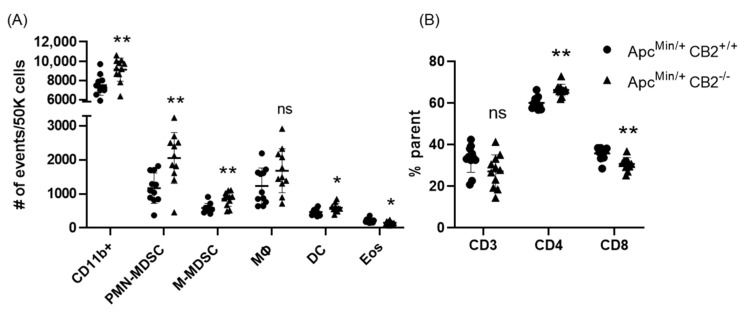
The CB2 receptor alters the balance between pro-tumor and anti-tumor cells in the spleen and tumor microenvironment of Apc^Min/+^ mice. (**A**) Relative frequency of CD11b+, CD11b+Ly6G^hi^Ly6C^int^ (PMN-MDSC), CD11b+Ly6G-Ly6C^hi^ (M-MDSC), CD11b+F480+ (macrophage, MΦ), CD11b^hi^Cd11c^hi^ (Dendritic cell, DC). The values are expressed as the number of events per 50,000 cells. (**B**) CD3+, CD3+CD4+, and CD3+CD8+ T cells in the spleen. CD3 expressed as the percent of single cells; CD4 and CD8 expressed as the percent of CD3 cells. Apc^Min/+^ CB2^+/+^, n = 11 (5 females and 6 males); Apc^Min/+^ CB2^−/−^, n = 11 (5 females and 6 males). Student’s *t*-test, * *p* < 0.05 and ** *p* < 0.01; ns: not significant.

**Figure 7 ijms-24-04060-f007:**
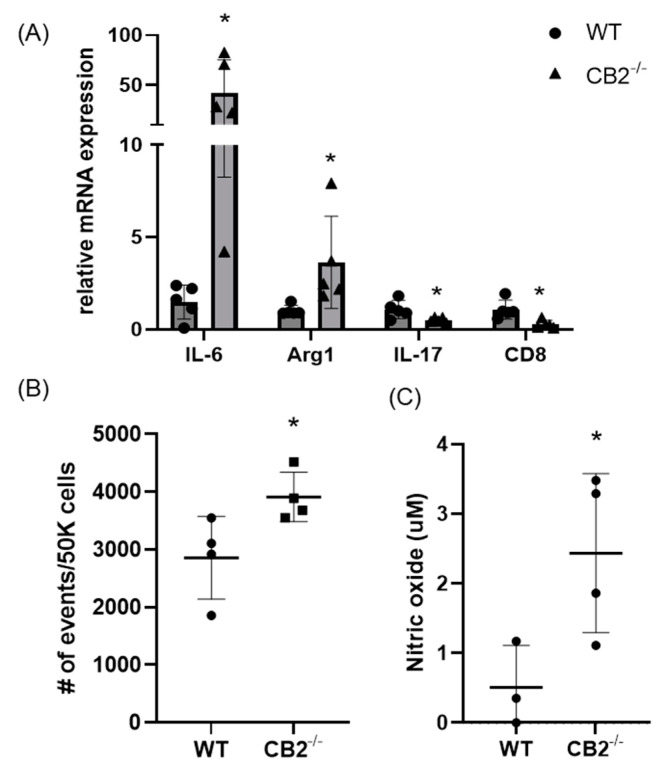
The CB2 receptor alters the balance between pro-tumor and anti-tumor cells and biomarkers in the TME in AOM/DSS-treated mice. (**A**) Relative mRNA expression of IL-6, arginase-1 (Arg1, functional marker of MDSCs), IL-17, and CD8 in the TME. (**B**) Relative frequency of PMN-MDSCs (CD11b+Ly6G+) in the TME. (**C**) Nitric oxide levels in the tumor microenvironment of polyps from the distal colon. WT, n ≥ 3; CB2^−/−^, n ≥ 3 as indicated, Student’s *t*-test, * *p* < 0.05.

**Figure 8 ijms-24-04060-f008:**
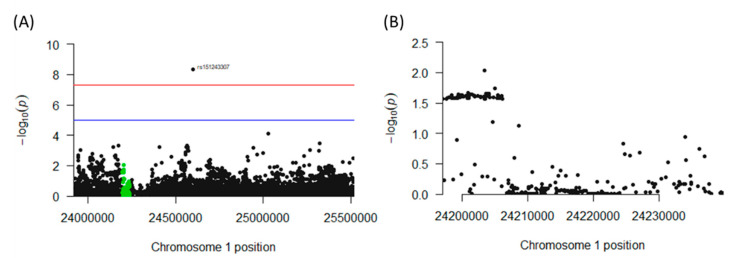
Association of polymorphisms in the *CNR2* gene with colon cancer incidence in humans. (**A**) Chromosome 1 haplotype block containing the *CNR2* gene. This genomic region houses the *CNR2* gene (highlighted in green) and all the SNPs found in the region. (**B**) Cross-section of the genomic region of the *CNR2* gene that contains an aggregate of significant (*p* < 0.05) SNPs associated with colon cancer incidence.

## Data Availability

The data were generated by the authors and are available upon request. The genomic study data, sources, and software used are included in the article and/or Appendix A.

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
