# Peer review of "The Anti-Tumorigenic Role of Cannabinoid Receptor 2 in Colon Cancer: A Study in Mice and Humans"

_ijms, 2023, doi:10.3390/ijms24044060_

Round 1

Reviewer 1 Report

This research article is designed to investigate a role of cannabinoid receptor-2 (CB2 or CNR2) in chemical (AOM)-induced formation of ACFs and Apc-mutation associated tumorigenesis using wild type and CB2 knockout mice. Authors also performed genomic analysis in a large human population to estimate a potential correlation between variation of CNR2 and the risk of colon cancer. Authors claim that CB2 knockout increased tumorigenesis in both mouse models with enhanced inflammation and there is a significant correlation between CNR2 variation and incidence of human colon cancer. I believe that this study will expand our understanding to a significant role of CB2 (CNR2) in cannabinoids (or any CB2/CNR2 ligands)-mediated anti-tumorigenic activity in colon cancer. The manuscript is overall well written and establishes very interesting finding.

I have a couple of minor points that need to be addressed.

More detailed information on crossbreeding of CB2 KO mice with ApcMin/+ mice, and total animal number (# of male and female) used for the experiments and analyses should be described in “Materials and Methods” and “Figure legends”.  

I was wondering if there is any speculation for gender disparity on a higher incidence of spontaneous precancerous lesions in CB2-/- female mice.

Author Response

More detailed information on crossbreeding of CB2 KO mice with ApcMin/+ mice, and total animal number (# of male and female) used for the experiments and analyses should be described in “Materials and Methods” and “Figure legends”.  

Response: An excerpt about crossbreeding has been added to the Materials and Methods section (Lines 356-361)

Sample size has been added to Materials and Methods and figure legends (Lines 347, 351, 362-368, figures 4 and 6)

I was wondering if there is any speculation for gender disparity on a higher incidence of spontaneous precancerous lesions in CB2-/- female mice.

Response: A more detailed speculation regarding the potential influences of sex disparity has been added to the discussion (Lines 290-302)

Reviewer 2 Report

The manuscript "The anti-tumorigenic role of cannabinoid receptor 2 in colon cancer: a study in mice and humans" is really interesting and up-to-date. The manuscript is written well, the Introduction, Materials and Methods as well as Results well. The experimental design is logic. The intersexual differences in many aspects of physiology and pathophysiology are of high relevance nowadays, pointing out a very important issue. The authors may discuss deeper the intersexual differences, even they are outlined in the discussion.

Author Response

The intersexual differences in many aspects of physiology and pathophysiology are of high relevance nowadays, pointing out a very important issue. The authors may discuss deeper the intersexual differences, even they are outlined in the discussion.

RESPONSE: A more detailed explanation regarding the potential influences of sex disparity has been added to the discussion (Lines 290-302)

Text added in the discussion:

CB2 may act by modulating the levels of IL-6 under low inflammatory conditions, while CB2-/- females, which have increased IL-6 levels (Figure 2) even under non-inflammatory conditions, are at a higher risk for colon cancer in the steady-state because of the IL-6-induced upregulation of MDSCs. This is opposed to CB2-/- males, who show no difference in IL-6 levels at steady-state compared to WT. CB2 agonists have been shown to modulate the levels of sex hormones such as progesterone and estradiol, which in turn can affect IL-6 and myeloid cell proliferation [43, 44]. Also, sex differences in the response to exogenous cannabinoids appear to be strongly influenced by ovarian hormones [45]. This possibly explains the sex disparities seen in the spontaneous cancer model, and when coupled with our results, suggest that sex differences in endocannabinoid tone determine differential sensitivity and susceptibility to tumorigenesis. In contrast, under highly inflammatory conditions CB2 knockout mice and humans with reduced CNR2 function display higher levels of IL-6 independently of sex [24, 25, 46, 47]. The observation that IL-6 is strongly upregulated in the inflammatory environment of ApcMin/+ mice [48, 49] and colon cancer patients in males and females [50-52], may therefore explain the lack of sex-related differences detected in the influence of CB2 on colon cancer severity in our ApcMin/+ model (Figure 4).